# Electronegative LDL Is Associated with Plaque Vulnerability in Patients with Ischemic Stroke and Carotid Atherosclerosis

**DOI:** 10.3390/antiox12020438

**Published:** 2023-02-10

**Authors:** Núria Puig, Pol Camps-Renom, Arnau Solé, Ana Aguilera-Simón, Elena Jiménez-Xarrié, Alejandro Fernández-León, Mercedes Camacho, Marina Guasch-Jiménez, Rebeca Marin, Joan Martí-Fàbregas, Alejandro Martínez-Domeño, Luis Prats-Sánchez, Francesca Casoni, Belén Pérez, Francesc Jiménez-Altayó, Jose Luis Sánchez-Quesada, Sonia Benitez

**Affiliations:** 1Cardiovascular Biochemistry, Institut d’Investigació Biomèdica Sant Pau (IIB SANT PAU), 08041 Barcelona, Spain; 2Department of Biochemistry and Molecular Biology, Faculty of Medicine, Universitat Autònoma de Barcelona, 08193 Barcelona, Spain; 3Stroke Unit, Department of Neurology, Hospital de La Santa Creu i Sant Pau, IIB SANT PAU, 08041 Barcelona, Spain; 4Department of Nuclear Medicine, Hospital de la Santa Creu i Sant Pau, IIB SANT PAU, 08041 Barcelona, Spain; 5Genetic of Complexes Diseases, Institut d’Investigació Biomèdica Sant Pau (IIB SANT PAU), 08041 Barcelona, Spain; 6Neurology-Sleep Disorders Center, Department of Clinical Neurosciences, IRCCS San Raffaele Scientific Institute, 20132 Milan, Italy; 7Therapeutics and Toxicology, Department of Pharmacology, School of Medicine, Universitat Autònoma de Barcelona, 08193 Barcelona, Spain; 8Neuroscience Institute, Universitat Autònoma de Barcelona, 08193 Barcelona, Spain; 9CIBER of Diabetes and Metabolic Diseases (CIBERDEM), Instituto de Salud Carlos III, 28029 Madrid, Spain

**Keywords:** electronegative LDL (LDL(−)), oxidized LDL (oxLDL), ischemic stroke, carotid atherosclerosis, plaque vulnerability

## Abstract

Owing to the high risk of recurrence, identifying indicators of carotid plaque vulnerability in atherothrombotic ischemic stroke is essential. In this study, we aimed to identify modified LDLs and antioxidant enzymes associated with plaque vulnerability in plasma from patients with a recent ischemic stroke and carotid atherosclerosis. Patients underwent an ultrasound, a CT-angiography, and an ^18^F-FDG PET. A blood sample was obtained from patients (n = 64, 57.8% with stenosis ≥50%) and healthy controls (n = 24). Compared to the controls, patients showed lower levels of total cholesterol, LDL cholesterol, HDL cholesterol, apolipoprotein B (apoB), apoA-I, apoA-II, and apoE, and higher levels of apoJ. Patients showed lower platelet-activating factor acetylhydrolase (PAF-AH) and paraoxonase-1 (PON-1) enzymatic activities in HDL, and higher plasma levels of oxidized LDL (oxLDL) and electronegative LDL (LDL(−)). The only difference between patients with stenosis ≥50% and <50% was the proportion of LDL(−). In a multivariable logistic regression analysis, the levels of LDL(−), but not of oxLDL, were independently associated with the degree of carotid stenosis (OR: 5.40, CI: 1.15–25.44, *p* < 0.033), the presence of hypoechoic plaque (OR: 7.52, CI: 1.26–44.83, *p* < 0.027), and of diffuse neovessels (OR: 10.77, CI: 1.21–95.93, *p* < 0.033), indicating that an increased proportion of LDL(−) is associated with vulnerable atherosclerotic plaque.

## 1. Introduction

Approximately 20% of all ischemic strokes are attributed to atherosclerosis, mainly in the internal carotid artery. The degree of stenosis is a well-recognized marker of carotid plaque instability and of the risk of recurrence, and currently determines the decision whether to perform carotid revascularization in most cases [1]. However, the degree of stenosis alone is not sufficient to make this decision in some frequent clinical situations [2]. Validated markers for monitoring vascular risk and the response to treatment of these patients are lacking. Therefore, the study of new imaging and plasma biomarkers that complement and improve the assessment of carotid plaque vulnerability in patients with ischemic stroke is essential and may set the basis for future clinical trials.

Imaging markers may predict the risk of recurrence beyond the degree of stenosis. Some examples include plaque echoluceny, intraplaque neovascularization, intraplaque hemorrhage, or inflammation assessed by positron emission tomography with ^18^F-fluorodeoxyglucose (^18^F-FDG PET) [3,4]. However, studies describing plasma biomarkers related to plaque vulnerability are still scarce. A recent study has revealed the increase of several plasma inflammatory markers in ischemic stroke patients with carotid atherosclerosis [5]. Among them, soluble intercellular adhesion molecule-1 (sICAM-1), soluble vascular adhesion molecule-1 (sVCAM-1), and fractalkine (FKN) were associated with carotid plaque inflammation evaluated by ^18^F-FDG PET. sICAM-1 was also associated with stroke recurrence [5]. Another important study demonstrated that baseline levels of IL-6 were associated with plaque vulnerability and progression [6].

Owing to their contribution to inflammation and oxidative stress, the role of lipoproteins, mainly modified forms of low-density lipoprotein (LDL), in the progression of atherosclerosis is widely established [7]. Several studies demonstrate that increased LDL cholesterol (LDLc) level influences vascular risk, including atherothrombotic stroke [8,9]. In contrast, high-density lipoprotein cholesterol (HDLc) levels are inversely associated with vascular risk and with stroke recurrence [10,11]. However, therapies that increase HDL or decrease LDL are not enough to eliminate vascular risk, a fact that reveals the need to look for biomarkers related to lipoprotein functionality. Beyond LDLc, the presence of modified LDL, such as oxidized LDL (oxLDL), in plasma and atherosclerotic plaque determines the susceptibility to developing vulnerable lesions [12,13]. Several studies have shown an association between circulating oxLDL and atherosclerotic plaque [12,14], and with the presence and prognosis of stroke [15,16]. Electronegative LDL (LDL(−)) is a minor plasma form of modified LDL with inflammatory properties whose proportion is increased in acute myocardial infarction [17], in acute ischemic stroke [18], and in pathologies associated with vascular risk, such as dyslipemias or diabetes [19], and is associated with the extent of carotid stenosis [20].

Unlike LDL, HDL exerts atheroprotective actions, such as the induction of cholesterol efflux (ChE) from tissues [21,22], and anti-oxidant and anti-inflammatory properties [23]. Two enzymatic activities contained in HDL contribute to these properties: platelet-activating factor acetylhydrolase (PAF-AH), also known as lipoprotein-associated phospholipase A2, and paraoxonase-1 (PON-1) [24,25]. The functionality of lipoproteins also relies on the content of lipids and apolipoproteins (apo). Therefore, we evaluate in serum from ischemic stroke patients with carotid atherosclerosis: (1) concentrations of lipids and apolipoproteins; (2) anti-oxidant and atheroprotective activities associated with HDL; (3) levels of modified LDL in circulation; and (4) the association of these parameters with features of carotid plaque vulnerability, evaluated by imaging techniques. Altogether, the aim of the present study was to assess which molecules are putative indicators of vulnerability features of the carotid plaque in ischemic stroke patients.

## 2. Materials and Methods

### 2.1. Study Design

An observational cohort study (NCT03218527) of consecutive adult patients who had had a recent anterior circulation ischemic stroke and carotid atherosclerosis was conducted in the Hospital Santa Creu i Sant Pau between January 2016–March 2019. A control group of healthy subjects was also included. The study was approved by the Ethics Committee of the Hospital (IIBSP-LRB-2017-54, 26 June 2017), and the patients or their legal representatives gave written consent to participate. The study was performed in accordance with the Helsinki Declaration.

### 2.2. Study Population

Patients were included in the study if they fulfilled the following criteria: (1) age ≥ 50 years; (2) anterior circulation ischemic stroke or transient ischemic attack (TIA) within 7 days before inclusion; (3) at least one atherosclerotic plaque in the internal carotid artery (ICA) on the side consistent with stroke symptoms, regardless of the degree of stenosis. Carotid stenosis was graded using the NASCET approach [26] with computed tomography (CT) or magnetic resonance imaging-angiography, as well as being based on hemodynamic criteria using ultrasound [27]; and (4) previous modified Rankin Scale (mRS) score <4. The exclusion criteria were: (1) presence of cardioembolic, lacunar, or unusual stroke etiology according to the TOAST criteria [28]; (2) presence of a hemodynamic stroke/TIA; (3) prior carotid surgery or stenting; (4) presence of comorbidities conditioning a life-expectancy <1 year; (5) concomitant infections at the time of blood extraction; and (6) total artery occlusion.

Healthy controls fulfilled the following criteria: (1) age ≥ 50 years, (2) no prior history of ischemic heart disease, and (3) no prior stroke.

The following clinical variables were recorded for all of the patients: (1) age and sex; (2) past medical history including hypertension, diabetes, dyslipidemia, prior stroke/TIA, coronary artery disease, tobacco, and alcohol consumption; (3) previous treatments; (4) National Institutes of Health Stroke Scale (NIHSS) score, as a surrogate of infarct size; (5) body mass index (BMI); (6) regular physical exercise according to the physician-based assessment and counseling for exercise (PACE) scale [29]; (7) Mediterranean diet adherence according to the PREDIMED score [30]; (8) mRS score at inclusion; (9) stroke etiology according to the TOAST criteria [28] after a diagnostic work-up that included at least a 24-h-electrocardiogram, an echocardiogram, and an ultrasound carotid examination (assessing plaque echolucency, plaque surface, and degree of stenosis by hemodynamic criteria); and (10) results from the admission blood test, including renal function, hemogram, hemostasis, and lipid profile. After the stroke, the treating clinicians provided medical and revascularization treatments according to guidelines [31].

All the healthy controls underwent a clinical interview to assess demographics, lifestyle habits, and prior treatments. Additionally, a standard B-mode and color-Doppler carotid ultrasound examination was performed to rule out the presence of asymptomatic carotid stenosis in both ICA and an electrocardiogram to rule out silent ischemic heart disease and atrial fibrillation. Healthy subjects did not undergo ^18^F-FDG PET examination.

### 2.3. Carotid Plaque Imaging

All stroke patients included in the study underwent a carotid ultrasound, a CT-angiography or MR-angiography, and an ^18^F-FDG PET/CT within 15 days from the index stroke.

The ultrasound protocol included a duplex examination and a Contrast-Enhanced Ultrasound Study (CEUS) to determine the presence of neovascularization. Inflammation of the carotid plaques was assessed by ^18^F-FDG PET according to the maximum Standardized Uptake Value (SUV) within the plaque. These imaging protocols are detailed below.

#### 2.3.1. Carotid Ultrasound Protocol

All of the patients underwent a carotid ultrasound (US) exam, including a Contrast-Enhanced Ultrasound (CEUS) study, within 15 days from the occurrence of the index stroke. An experienced examiner, P.C.-R., certified in Neurosonology by the Spanish Society of Neurology, performed the US examinations using a Philips CX50^®^ Ultrasound Machine (Philips, Amsterdam, Netherlands) with a linear probe. The US study protocol consisted of two parts: a standard B-mode and color-Doppler carotid plaque characterization, and the CEUS examination.

The extracranial common carotid arteries and the ICAs were examined in the longitudinal and the transverse planes. A plaque was defined as a localized lumen narrowing of ≥1.5 mm or an increase of >50% in the intima-media thickness compared to the adjacent portion of the vessel wall. When a plaque was identified, the following sonographic variables were recorded: (1) morphology of the plaque (concentric or eccentric); (2) echogenicity of the plaque classified as I = uniformly hypoechoic, II = predominantly hypoechoic, III = predominantly hyperechoic, IV = uniformly hyperechoic; and V = calcified plaque [32]; and (3) degree of stenosis by hemodynamic criteria [27]. Only the largest plaque was studied when patients presented more than one plaque in the ICA. For the statistical analysis, the echogenicity of the plaque was also classified as predominantly hypoechoic (<50% of the surface, comprising I and II categories) and predominantly hyperechoic (≥50% of the surface, comprising III, IV, and V categories).

After the B-mode and color-Doppler characterization, the CEUS examination was performed using the preset real-time, contrast-enhanced imaging modality with coded pulse inversion from the Philips CX50^®^ Ultrasound Machine. This setting decreases the mechanical index to 0.1, obtaining an almost completely black screen in the absence of contrast. Then, a bolus of 2 mL of Sonovue^®^ contrast was injected into a peripheral vein and flushed with 10 mL of saline according to the recommendations of the manufacturer (Bracco Imaging, Milan, Italy). At that point, the lumen was filled with the hyperechoic bubbles of the contrast defining the perimeter of the plaque in negative. Time gain compensation was adjusted to achieve homogeneous signal intensity. Finally, a DICOM cine loop was recorded for 120 s starting when the contrast bolus was injected and plaque neovessels were identified as hyperechoic bubbles appearing within the plaque perimeter.

The neovascularization of plaque was classified into grades: 0 (no visible microbubbles within the plaque), 1 (moderate microbubbles confined to the shoulder and/or adventitial side of the plaque), and 2 (diffuse microbubbles throughout the plaque), as previously described [33]. This grading was performed by P.C.-R. and by F.C., who was blinded to all of the clinical information, to calculate the interrater agreement and the Cohen’s kappa coefficient. When a discrepancy was detected, the images were reviewed, and a consensus between raters was required. A plaque was suitable for analysis of neovascularization if a DICOM cine loop of 15 s presented enough quality without movement artifact (for example swallowing) and at least 50% of the plaque was visible without calcium shadows.

#### 2.3.2. Carotid ^18^F-FDG PET/CT Protocol

Carotid ^18^F-FDG PET was performed in a Philips Gemini TF TOF 64 PET/CT (Philips Medical System, Eindhoven, The Netherland). The examinations were performed after a fast that lasted a minimum of six hours. PET scans were not performed if pre-PET blood glucose exceeded 10 mmol/L. Two hours before image acquisition, 320 MBq of ^18^F-FDG was administered. The uptake phase was standardized with the patient resting. PET images were acquired in a 3-dimensional mode in 2-bed positions for 10 min each. Images from CT angiography and PET were co-registered afterwards to assess the slice of maximal plaque stenosis. 

^18^F-FDG activity was measured in 10 regions of interest, which were defined relative to the slice of maximal stenosis on the co-registered CT angiography, corresponding to a 1 mm axial plaque slice (5 distal and 5 proximal). ^18^F-FDG was quantified using standardized uptake values (SUV g/mL, defined as measured uptake [MBq/mL]/injected dose [MBq] per patient weight [g]). We defined the single hottest slice as the axial slice with maximal SUV uptake (SUVmax) [4].

### 2.4. Collection of Blood Samples

Peripheral blood samples from the stroke patient were collected on day 7 ± 1 from the stroke. Plasma was collected in ethylenediaminetetraacetic acid (EDTA)-containing Vacutainers and serum in Serum Separator Tubes with clot activator. The tubes were centrifuged at 1500× *g* for 15 min at 4 °C, and the aliquots were frozen at −80 °C.

### 2.5. Serum Determinations

The lipid profile and apolipoprotein concentrations included total cholesterol, triglycerides, non-esterified fatty acids (NEFAs), apoB, apoA-I, apoA-II, apoC-III, apoE, apoJ, very-low-density lipoprotein cholesterol (VLDLc), LDLc, HDLc, and oxLDL. The cholesterol of lipoprotein fractions was quantified by using a direct HDLc method (HDL-C plus, Abbott Core Laboratory, Chicago, IL, USA). Total cholesterol, triglycerides, apoB, and HDLc kits were from Abbott, and were measured in an Alinity ci-series autoanalyzer (Abbott). NEFA (Wako Chemicals, Osaka, Japan), apoA-I (Roche, Basel, Switzerland), apoA-II, apoC-III, and apoE (Kamiya Biomedical, Seattle, WA, USA) were measured in a Cobas 6000/c501 autoanalyzer (Roche). ApoJ and oxLDL were quantified by ELISA kits (Mabtech, Stockholm, Sweden and R&D Systems, Minneapolis, MN, USA, respectively) according to manufacturer’s instructions.

Parameters related to oxidation, malondialdehyde (MDA) levels and antioxidant capacity, were measured by the MDA/thiobarbituric test and by the 2,2-diphenyl-1-picrylhydrazyl (DPPH) test, respectively. Nitrite levels in serum were measured by the Griess method.

PAF-AH activity was quantified using 2-thio-PAF (Cayman Chemicals, Denver, CO, USA) as a substrate [34] and PON-1 activity by using phenylacetate (Sigma/Merck, Darmstadt, Germany), as described [35]. Total PAF-AH activity was measured in serum. In addition, both PAF-AH and PON-1 activities, as well as ChE, detailed hereafter, were also determined in apoB-depleted serum (HDL fraction) obtained by serum precipitation with dextran sulfate [36].

### 2.6. LDL(−) Quantification

Total LDL from stroke patients and controls was isolated from plasma by sequential flotation ultracentrifugation (density = 1.019–1.063 g/mL) and dialyzed against buffer A (Tris 10 mM, EDTA 1 mM, pH = 7.4). LDL(−) was isolated from 40 μg of total LDL and its proportion was quantified by using stepwise anion-exchange chromatography in a MonoQ 5/50 GL column (GE Healthcare, Chicago, IL, USA), as described [37].

### 2.7. Effect of Lipoproteins on Cells

Cell lines THP1-XBlue-MD2-CD14 monocytes (THP1-CD14) (Invivogen, San Diego, CA, USA) and endothelial cells Human Primary Coronary Artery Endothelial Cells (HCAEC) (ATCC, Manassas, VA, USA) were grown following the manufacturer’s recommendations. The growth medium for THP1-CD14 was RPMI 1640 supplemented with 10% fetal bovine serum (FBS) and, 1% Penicillin-Streptomycin from Biowest (Nuaille, France) and with NormocinTM (50 mg/mL), ZeocinTM (100 mg/mL), and G418 (100 mg/mL) antibiotics from Invivogen (San Diego, CA, USA). The growth medium for HCAEC was Vascular Cell Basal Medium, supplemented with the Endothelial Cell Growth Kit VEGF (ATCC, Manassas, VA, USA) which contains 2% of FBS. The experiments were performed as described below.

#### 2.7.1. ChE Capacity of HDL

The ChE promoted by 1% of apoB-depleted serum from patients and controls after 24 h’ incubation was determined in THP1-CD14 macrophages using fluorescent-labeled cholesterol, as described [38].

Briefly, THP1-CD14 monocytes were seeded at 200,000 cells/well with RPMI growth medium supplemented with phorbol 12-myristate 13-acetate (PMA) at 50 µg/L for 24 h. Afterwards, cholesterol (0.125 mM) and 20% of fluorescent cholesterol, topFluor-cholesterol linked to boron dipyrromethene (Avanti Polar Lipids, Alabaster, AL, USA), were added for 1 h. Then, cells were pre-treated with the liver X receptor agonist T0901317 (4 μmol/L) (Cayman Chemicals) for 18 h to stimulate efflux pathways. Macrophages were then incubated with 1% of apoB-depleted serum from patients and controls for 24 h. Finally, in order to calculate the efflux capacity, the fluorescence (λEx/Em = 485/530 nm) was measured by using a Synergy HT spectrophotometer (BioTek, Winooski, VT, USA) in both the supernatant and cells.

#### 2.7.2. Inflammatory Effect of LDL(−) on Cells

Total LDL was isolated from pooled plasma from normolipidemic volunteers by ultracentrifugation, and was separated in native LDL (LDL(+)) and LDL(−) by anion-exchange chromatography in an AKTA-FPLC system (GE Healthcare, Chicago, IL, USA).

THP1-CD14 monocytes were seeded (400,000 cells/well) with growth medium supplemented with PMA at 50 µg/L for 24 h to induce the differentiation into macrophages. HCAEC were seeded (50,000 cells/well) with growth medium for 24 h previously to the experiments. THP1-CD14 macrophages and HCAEC were incubated in the presence or absence of LDL(−) and LDL(+) at 60 mg apoB/L for 24 h with RPMI 1640 medium supplemented by 1% FBS and 1% Penicillin-Streptomycin (THP1-CD14) and with Vascular Cell Basal Medium supplemented by the Endothelial Cell Growth Kit VEGF without FBS (HCAEC). After incubation, cell supernatants were collected, and sICAM−1, sVCAM−1, and FKN release were evaluated by ELISA (R&D Systems, Minneapolis, MN, USA).

### 2.8. Outcomes

The primary outcome was finding alterations in the concentrations and functionality of lipoproteins and lipoprotein-related molecules in serum/plasma from ischemic stroke patients versus controls.

Secondary outcomes included the association of altered parameters with vulnerability features of the carotid plaque, including degree of stenosis, echolucency, neovascularization, and inflammation.

### 2.9. Statistical Analysis 

Continuous descriptive variables were reported as means and standard deviations (SD) or medians (md) and interquartile ranges (IQR) if they were not normally distributed. Categorical variables were expressed as counts and percentages. Differences between groups of patients (more than 2 groups) were assessed using Kruskal–Wallis rank-sum test. Differences on clinical characteristics and lipid parameters were analyzed among the stroke population, divided into two groups < 50% stenosis and ≥50% stenosis, and between all the stroke patients and the control group using the Student’s *t*-test or the Wilcoxon rank-sum test (when a non-parametric test was required) for continuous variables, and the χ^2^ test for categorical variables. Continuous variables not-normally distributed as assessed by the Shapiro–Wilk test were log-transformed to approach normality.

Correlations between parameters were analyzed using the Spearman’s correlation test. To estimate the association between the proportion of LDL(−) and the degree of plaque stenosis and other plaque features, a multivariable logistic regression analysis was performed using a backward stepwise selection modeling approach. Receiver Operating Characteristic (ROC) curve analyses were performed to compare the predictive value between LDL(−) and oxLDL.

Statistical significance for all the analyses was set at *p* < 0.05 (two-sided). Analyses were performed using Stata v.15 (StataCorp, College Station, TX, USA).

## 3. Results

### 3.1. Study Population

The study population included 37 patients with at least one atherosclerotic plaque causing ≥50% of stenosis, 27 patients with carotid stenosis <50%, and 27 healthy controls. The clinical characteristics of these groups are detailed in Appendix A. The groups were balanced for demographics and lifestyle habits except for current smoking and exercise, which were, respectively, more and less frequent in the stroke patients. While the patients’ BMI was lower, the presence of diabetes and treatment with antiplatelet agents and statins were higher in the stroke patients. In the ≥50% stenosis group, hypertension was less frequent than in the <50% group.

### 3.2. Lipid Profile and Lipoprotein-Related Molecules 

Table 1 shows the lipid profile and lipid-related parameters in plasma from stroke patients and from healthy patients. No changes were found according to the degree of stenosis; however, differences existed between all patients and controls. Patients showed decreased levels of total cholesterol, LDLc, and HDLc. Accordingly, they showed a decrease in the main apolipoproteins of LDL and HDL, apoB and apoA-I, respectively, and in the minor apolipoproteins apoA-II and apoE, but no significant differences in apoC-III. Serum from patients showed higher apoJ concentration and lower PAF-AH activity than that from healthy subjects.

No difference was found between groups, either in MDA or antioxidant capacity (Appendix A). No difference in the nitrite levels in serum of patients compared to Controls was observed (Appendix A). However, the achievement of conclusive results is hindered by the fact that these parameters were not evaluated in all the individuals because of lack of serum sample.

### 3.3. Anti-Atherogenic Properties of HDL

ApoB-depleted serum was used to analyze the protective properties of HDL. PAF-AH and PON-1 enzymatic activities, both related to the anti-oxidant and anti-inflammatory action of HDL, and the ability of apoB-depleted serum fraction to promote ChE from macrophages was evaluated. HDL-associated PAF-AH activity, PON-1 activity, and ChE were lower in stroke patients than in control subjects, although only PAF-AH and PON-1 reached statistical significance (Figure 1). No statistically significant differences between degree of stenosis <50% and ≥50% were found in these anti-atherogenic properties of HDL (Appendix A). No correlation was found between levels of HDLc and PAF-AH activity (Spearman’s rho = 0.14; *p* = 0.219) and ChE (Spearman’s rho = 0.17; *p* = 0.220), whereas there was a strong correlation between HDLc and PON-1 activity (Spearman’s rho = 0.48; *p* < 0.001). The apoJ/PON-1 ratio was increased in patients vs. controls (median apoJ/PON-1 activity = 5.24 [IQR 4.30–6.42] vs. 4.02 [IQR 3.11–4.65]; *p* < 0.001).

### 3.4. Modified LDL: oxLDL and LDL(−)

The decrease in the anti-oxidant enzymatic activities in HDL and the increase in the apoJ/PON-1 ratio suggested increased oxidative stress, which could favor higher oxidation of LDL in patients. Plasma oxLDL levels were higher in patients than in controls (median of oxLDL = 12.2 [IQR 7.8–17.3] vs. 7.4 [IQR 4.7–16.6], respectively; *p* = 0.042) (Figure 2a). Likewise, the percentage of plasma LDL(−) was significantly elevated in patients compared to controls (median % of LDL(−) = 7.9 [IQR 6.2–10.2] vs. 6.5 [IQR 4.1–7.9], respectively; *p* = 0.004) (Figure 2b). We did not find any correlation of LDL(−) with parameters of Table 1, with oxLDL, or with the inflammatory molecules associated with plaque inflammation [5] (Appendix A). However, oxLDL correlated positively with apoJ and the apoJ/PON-1 ratio, and negatively with apoC-III and PAF-AH, both in serum and associated with HDL (Appendix A).

There was no difference in oxLDL levels according to degree of stenosis (Figure 3a). By contrast, in patients with ≥50% of plaque stenosis, the proportion of LDL(−) was significantly higher than in controls and in <50% patients (*p* = 0.0035 and *p* = 0.0392, respectively). Therefore, the proportion of LDL(−) stratifying by the degree of stenosis was quantified (Figure 3b). Interestingly, for LDL(−), significance was already observed in patients with moderate stenosis (50–69%) compared to subjects without stenosis (controls) and patients with less than 50% stenosis.

### 3.5. Association of Plaque Characteristics with Modified LDL

Besides stenosis degree, we aimed to study the association between the proportion of LDL(−) and other features of plaque vulnerability, including echolucency, intraplaque neovascularization, and inflammation assessed by ^18^F-FDG PET/CT. Table 2 shows the levels of LDL(−), oxLDL, and LDLc by those characteristics.

The proportion of LDL(−) was significantly higher in those patients presenting predominantly hypoechoic plaques and intraplaque neovascularization, whilst LDLc and oxLDL did not show association with any of the plaque features analyzed.

Multivariable logistic regression analysis was performed to assess the association of LDL(−) with the characteristics of plaque vulnerability. Table 3 shows that LDL(−) was independently associated with the probability of presenting carotid stenosis ≥50% (a), predominantly hypoechoic plaque (b), and diffuse intraplaque neovascularization (c). Appendix A detail the bivariate analyses of predictors of carotid stenosis ≥50%, predominantly hypoechoic plaque and diffuse intraplaque neovascularization, respectively.

### 3.6. Prediction of Carotid Plaque Vulnerability According to the Plasma Proportion of LDL(−)

We performed ROC analyses to identify a balanced cut-off point of LDL(−) for predicting carotid plaque vulnerability with high sensitivity. We observed that a cut-off point of 6.9% of LDL(−) predicted the risk of presenting carotid stenosis ≥50% with a sensitivity of 82.9% and a specificity of 51.9%, predominantly hypoechoic plaque with a sensitivity of 94.4% and a sensitivity of 43.2%, and diffuse intraplaque neovascularization with a sensitivity of 85.7% and a sensitivity of 43.5%. Figure 4 shows that LDL(−), but not oxLDL, was a good predictor of such plaque features. By using this cut-off point in the multivariable logistic regression analyses (adjusted by the same variables as in Table 3), the presence of ≥6.9% of LDL(−) was independently associated with the presence of carotid stenosis ≥50% (OR = 4.98, 95% CI 1.46–16.97, *p* = 0.010) and of predominantly hypoechoic plaque (OR = 18.8, 95% CI 2.10–68.40; *p* = 0.009), and was almost significantly associated with intraplaque neovascularization (OR = 4.62, 95% CI 0.84–25.49; *p* = 0.079).

### 3.7. LDL(−) as an Inductor of Inflammation in Cultured Cells

Based on our previous findings of the association of sICAM-1 sVCAM-1 and FKN with plaque inflammation, we assessed whether LDL(−) induced the release of these molecules in cultured macrophages and endothelial cells. Figure 5 shows that LDL(−) did promote that induction, except for FKN in macrophages, which was undetectable.

## 4. Discussion

In the present study, patients with a recent ischemic stroke and carotid atherosclerosis showed increased blood levels of oxLDL and LDL(−). The proportion of LDL(−), but not that of oxLDL, was associated with the degree of carotid stenosis, as well as with other features of plaque vulnerability, such as echolucency and neovascularization. In addition, we demonstrated that LDL(−) induced the release of sICAM-1, sVCAM-1, and FKN in endothelial cells and macrophages.

It is widely accepted that the main inductor of inflammation and lipid accumulation in atherosclerotic lesions is modified LDL retained in the intima [7]. Therein, one of the main sources of LDL modification is the oxidative milieu yielded by radical oxygen species (ROS) and oxidative enzymes released by cells from the arterial wall. In this study, oxLDL and LDL(−) were higher in the blood from patients with a recent ischemic stroke and carotid atherosclerosis than in healthy controls. Contrariwise, the patients showed a more favorable lipid profile than control subjects, having lower levels of total cholesterol and LDLc. This may be attributable to the known decrease in lipid levels after ischemia and to the administration of statin immediately after the ischemic stroke. In line with the decrease in LDL and HDL, the main apolipoproteins in them (apoB, apoA-I, and apoA-II, respectively) and minor apoprotein apoE were also lower in patients. Otherwise, the levels of apoJ were higher, as was the apoJ/PON−1 ratio, and both correlated with oxLDL levels. Accordingly, the apoJ/PON−1 ratio was shown to be elevated in patients with cardiovascular risk and in atherosclerotic animal models, as well as in the presence of oxLDL [39]. Increased apoJ levels in the plasma of ischemic patients could be interpreted as a response to ischemia, since apoJ is known to act as an acute phase protein [40].

The protective action of HDL was altered in patients, as is suggested by its lower PAF-AH and PON-1 enzymatic activities, and by the trend to lower ChE capacity. This diminution may be due not only to the lower amount of HDL particles but also to a lack of HDL functionality in patients. The decrease in PON-1 and PAF-AH activities is closely involved in the anti-oxidant and anti-inflammatory properties ascribed to HDL [25], which may contribute to the inflammatory state of patients favoring LDL oxidation. In this regard, the activity of PAF-AH in serum and of PAF-AH associated with HDL correlated with oxLDL levels. Although higher concentrations of oxLDL in serum were found in the stroke patients of our study, no difference was observed in its levels depending on the degree of stenosis or other features of plaque vulnerability.

The most significant findings of the current study are related to the predictive value of plasma LDL(−) for plaque vulnerability. To the best of our knowledge, only one previous study had reported increased levels of LDL(−) in ischemic stroke patients [18], despite showing higher proportions than in our study, presumably as a result of methodological differences, mainly the type of patients, the time of blood extraction, and the chromatographic method of LDL(−) isolation. Our main highlights are that: (1) Stroke patients had a higher proportion of LDL(−), independently of other lipid-related molecules and of treatment with high-dose statins [41,42,43]; and (2) LDL(−), but not oxLDL, was independently associated with the degree of carotid stenosis, hypoechogenicity, and diffuse intraplaque neovascularization, all of which are features of plaque vulnerability. The ability of LDL(−) to induce inflammation and foam cell formation [19,44], as well as the expression of molecules related to angiogenesis in endothelial cells and macrophages [44,45], may underlie these characteristics of carotid plaque. Moreover, LDL(−) shows a high affinity to proteoglycans of the artery wall [46] and hence it is prone to be retained in the subendothelial space, where it is able to interact with cell types present in the atherosclerotic lesion. In addition, LDL(−) promotes the release of MMP [47], a fact that is likely contributing to enhanced plaque vulnerability.

In our study there was no correlation between LDL(−) and oxLDL levels, and there was neither higher MDA concentration nor antioxidant capacity in serum from patients. In agreement, our previous observations showed no evidence of oxidative modification in LDL(−) compared to native LDL, since they have a similar content of MDA, fatty acid hydroxides, and antioxidants [48]. Previous literature has reported that LDL(−) elicits effects related to oxidative stress in endothelial cells, such as mitochondrial free-radical production and senescence [49], superoxide dismutase expression [50], and inhibition of endothelial nitric oxide synthase activation [51,52]. Notably, in a very recent study conducted in the same cohort of patients, we found that serum from ischemic patients with a high degree (≥70%) of stenosis promoted cyclooxygenase-dependent endothelial dysfunction in carotid arteries of non-ischemic mice [53]. Therefore, the relationship among LDL(−), oxidative stress, and endothelial dysfunction in atherothrombotic stroke patients deserves to be elucidated and is being evaluated in our ongoing investigations.

In another recent study, we described how in the same cohort of patients there was an association between carotid plaque inflammation and sICAM-1, sVCAM-1, and FKN concentrations in plasma. Interestingly, the in vitro experiments of the present study showed that LDL(−) promote the release of those molecules in endothelial cells and in macrophages, likely contributing to the inflammatory state within the plaque and to the presence of neovessels, since such molecules are related to angiogenesis [54,55,56]. We hypothesize, as illustrated in Figure 6, that LDL(−) may be more a cause than a consequence of plaque progression, and hence it mainly acts earlier by promoting plaque vulnerability, but without showing a direct association with the plasma inflammatory molecules released at the onset of stroke and associated with SUVmax [5]. This does not eliminate the possibility that a part of plasma LDL(−) may also be generated by lipolysis/proteolysis in the arterial wall and then released from vulnerable plaque.

The present study has some limitations. As the number of patients was small, evaluation in larger cohorts is required to confirm the findings. In addition, some unresolved questions emerge from the present results, for example regarding the evaluation of parameters related to lipid oxidation and antioxidant capacity in plasma. In order to answer those questions, ongoing studies are being conducted in the same cohort of patients. This investigation is mainly focused on determining the oxidative parameters in plasma, as well as the molecular composition of the isolated lipoproteins, and the oxidative properties of LDL and the anti-oxidative protection of HDL against LDL oxidation.

## 5. Conclusions

In conclusion, ischemic stroke patients with carotid atherosclerosis, in spite of treatment with statin and a favorable lipid profile, showed high levels of LDL(−). This observation, in conjunction with the association with the degree of stenosis, plaque neovascularization, and hypoechogenicity, suggests that LDL(−) is a much better indicator of plaque vulnerability than oxLDL. In addition to the increased presence of inflammatory LDL(−), the lower levels of HDL with less protective enzymes may converge in an enhanced inflammatory response in these patients. Although further investigation is required to assess its predictive usefulness, our findings suggest that LDL(−) may be useful in the future for the development of therapeutic strategies, in symptomatic and also in asymptomatic patients with carotid atherosclerosis.

## Figures and Tables

**Figure 1 antioxidants-12-00438-f001:**
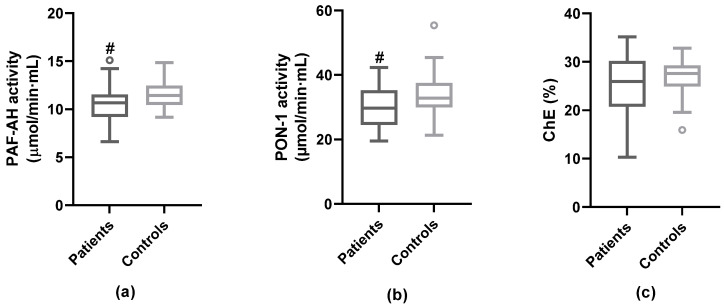
Parameters of HDL functionality. PAF-AH and PON-1 activities and ChE were quantified in apoB-depleted serum. PAF-AH and PON-1 activities were calculated from the slope of the enzymatic activity measured by colorimetric detection. (**a**) PAF-AH activity was determined by using 2-thio-PAF as a substrate. (**b**) PON-1 activity was measured using phenylacetate as a substrate. (**c**) ChE capacity of apoB-depleted serum was analyzed in THP1-CD14 macrophages after 24 h incubation. Patients (n = 46); Controls (n = 26); # vs. Controls; *p* < 0.05.

**Figure 2 antioxidants-12-00438-f002:**
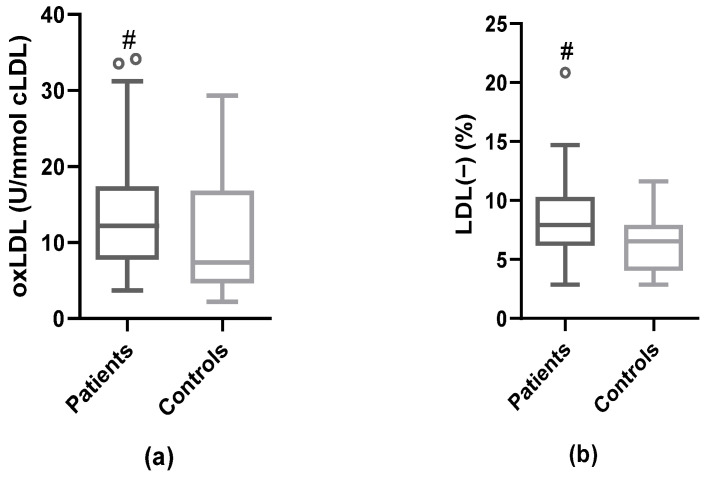
Concentration of oxLDL and LDL(−) levels compared to controls. (**a**) Levels of serum oxLDL measured by ELISA in patients and controls. (**b**) Plasma LDL(−) proportion quantified by anion-exchange chromatography in patients and controls. # vs. controls; *p* < 0.05.

**Figure 3 antioxidants-12-00438-f003:**
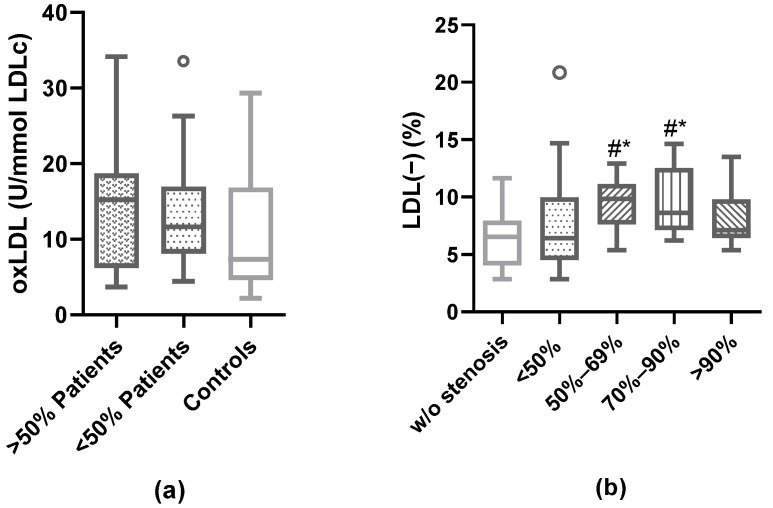
Concentration of oxLDL and LDL(−) according to stenosis degree. Serum oxLDL levels were measured by ELISA and plasma LDL(−) proportion was quantified by anion-exchange chromatography. (**a**) Levels of oxLDL when the patients were divided into groups of >50% stenosis and <50% stenosis. (**b**) Proportion of LDL(−) stratifying by degree of carotid stenosis (*w*/*o* stenosis n = 27, <50% n = 27, 50– 69% n = 13, 70–90% n = 12, >90% n = 10), Kruskal–Wallis test was performed (*p* = 0.0017), # vs. controls (w/o stenosis); * vs. <50% patients; *p* < 0.05.

**Figure 4 antioxidants-12-00438-f004:**
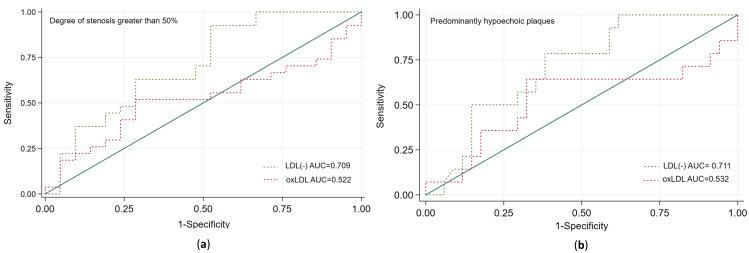
ROC comparison analyses between LDL(−) and oxLDL. (**a**) Prediction of a carotid degree of stenosis greater than 50%. (**b**) Prediction of predominantly hypoechoic plaque.

**Figure 5 antioxidants-12-00438-f005:**
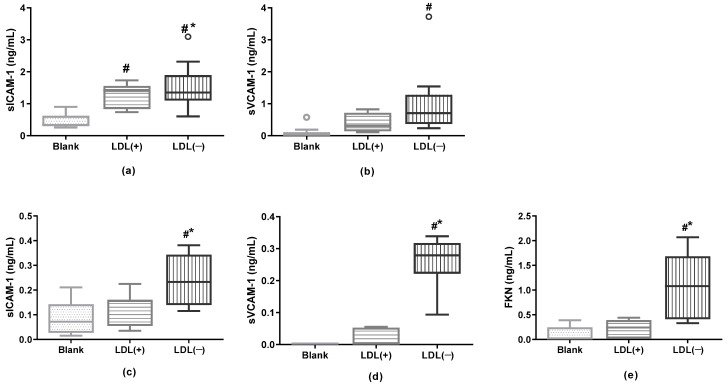
LDL(−)-induced release of inflammatory molecules in macrophages and endothelial cells. THP1-CD14 macrophages (400,000 cells/well) and HCAEC (50,000 cells/well) were incubated in the presence or absence of LDLs (60 mg apoB/L) for 24 h. sICAM-1, sVCAM-1, and FKN release were then evaluated by ELISA in the supernatant of (**a**,**b**) THP1-CD14 macrophages (n = 11) and of (**c**–**e**) HCAEC (n = 7). The Kruskal–Wallis test was performed (*p* < 0.01 in all cases), # vs. Blank; * vs. LDL(+); *p* < 0.05.

**Figure 6 antioxidants-12-00438-f006:**
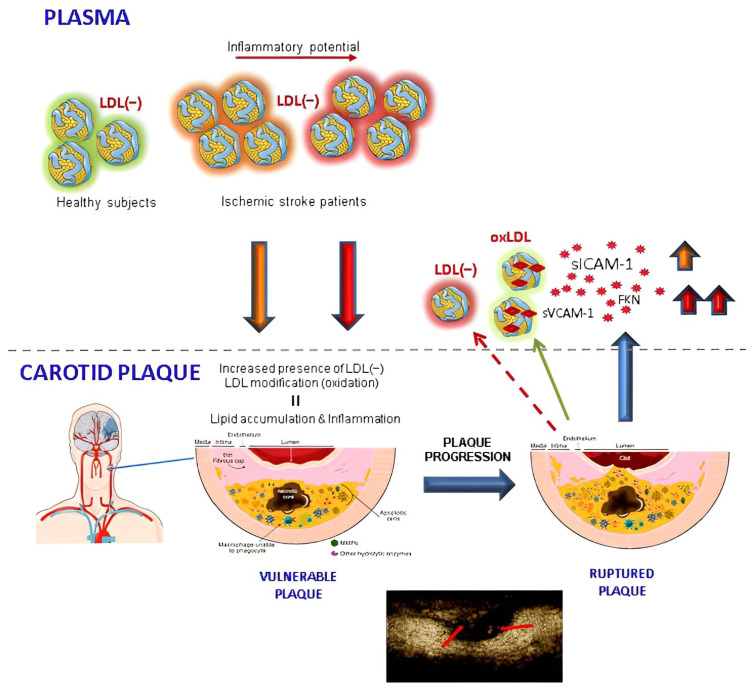
Role of LDL(−) in ischemic stroke associated with atherosclerosis. Before the stroke event, subjects with a high susceptibility to developing carotid atherosclerosis show a high percentage of LDL(−) with inflammatory properties. At first, LDL(−) contributes to carotid plaque formation and promotes the progress of the lesion by the sustained induction of inflammation and lipid accumulation. Then, as a result of the progression of atherosclerosis, the plaque ruptures and releases part of its inflammatory content and modified LDLs into the circulation, eventually leading to the formation of thrombus and a stroke event.

**Table 1 antioxidants-12-00438-t001:** Concentration of lipid profile and lipoprotein-related parameters according to the study population.

	<50% Group (n = 27)	≥50% Group (n = 37)	*p*	All Stroke Patients (n = 64)	Control Group (n = 27)	*p*
Triglycerides (mM), md (IQR)	1.11 (0.88–1.53)	1.20 (0.94–1.63)	0.601	1.17 (0.92–1.61)	1.18 (0.90–1.45)	0.212
Total cholesterol (mM), m ± sd/md (IQR)	3.80 ± 1.03	3.95 ± 1.186	0.588	3.75 (2.91–4.65)	4.64 (4.00–5.88)	**<0.001**
VLDLc (mM), md (IQR)	0.22 (0.18–0.31)	0.24 (0.19–0.33)	0.601	0.23 (0.18–0.32)	0.24 (0.18–0.29)	0.524
LDLc (mM), md (IQR)	2.48 (1.76–3.48)	2.31 (1.88–3.20)	0.949	2.41 (1.82–3.29)	3.09 (2.58–3.91)	**<0.001**
HDLc (mM), m ± sd/md (IQR)	1.07 ± 0.37	1.02 ± 0.28	0.541	1.03 (0.80–1.32)	1.34 (1.15–1.59)	**<0.001**
LDLc/HDLc ratio, md (IQR)	2.42 (1.85–2.93)	2.57 (1.98–3.11)	0.510	2.48 (1.92–3.03)	2.25 (1.98–2.80)	0.507
NEFA (mM), md (IQR)	0.47 (0.30–0.69)	0.38 (0.26–0.57)	0.404	0.42 (0.27–0.61)	0.43 (0.24–0.48)	0.621
apoB (g/L), md (IQR)	0.68 (0.56–0.90)	0.66 (0.60–0.80)	0.826	0.68 (0.58–0.83)	0.91 (0.78–1.02)	**<0.001**
apoA-I (g/L), md (IQR)	1.22 (1.14–1.50)	1.22 (1.10–1.47)	0.703	1.22 (1.10–1.47)	1.62 (1.44–1.87)	**<0.001**
apoA-II (g/L), m ± sd	0.30 ± 0.07	0.28 ± 0.08	0.468	0.30 ± 0.08	0.37 ± 0.08	**<0.001**
apoE (g/L), m ± sd	0.04 ± 0.02	0.04 ± 0.02	0.978	0.04 ± 0.02	0.05 ± 0.02	**0.006**
apoC-III (g/L), md (IQR)	0.05 (0.03–0.09)	0.05 (0.01–0.11)	0.713	0.05 (0.02–0.10)	0.07 (0.03–0.11)	0.144
apoJ (mg/L), md (IQR)	170 (153–219)	180 (147–218)	0.952	179 (148–218)	145 (114–170)	**<0.001**
Total PAF-AH activity (µmol/min*mL) m ± sd	19.10 ± 3.33	19.07 ± 3.58	0.974	19.08 ± 3.41	21.49 ± 2.66	**0.003**

VLDLc (very low-density lipoprotein cholesterol); LDLc (low-density lipoprotein cholesterol); HDLc (high-density lipoprotein cholesterol); NEFA (non-esterified fatty acid); apo (apolipoprotein); PAF-AH (platelet-activating factor acetylhydrolases). Student’s *t*-test or Wilcoxon rank-sum test (when a non-parametric test was required) were used to compare groups; *p* < 0.05 indicates significant differences.

**Table 2 antioxidants-12-00438-t002:** LDL(−), oxLDL, and LDLc levels by characteristics of plaque vulnerability.

Echolucency	Predominantly Hypoechoic (n = 19)	Predominantly Hyperechoic (n = 45)	*p*
LDL(−) (%), md (IQR)	9.5 (7.1–11.7)	7.4 (5.6–9.9)	**0.010**
oxLDL (U/mmol LDLc), md (IQR)	15.7 (5.4–21.6)	11.4 (8.2–17.1)	0.734
LDLc (mM), md (IQR)	3.02 (1.85–3.32)	2.33 (1.76–2.90)	0.228
*Intraplaque neovascularization*	**Present (n = 28)**	**Absent (n = 9)**	*p*
LDL(−) (%), md (IQR)	7.9 (6.5–10.5)	6.5 (3.8–7.6)	**0.047**
oxLDL (U/mmol LDLc), md (IQR)	13.2 (7.0–17.1)	12.3 (11.6–13.1)	0.976
LDLc (mM), md (IQR)	2.35 (1.89–3.20)	2.71 (2.01–3.70)	0.481
*Diffuse intraplaque neovascularization*	**Present (n = 14)**	**Absent (n = 23)**	*p*
LDL(−) (%), md (IQR)	9.0 (7.1–11.7)	7.0 (4.6–9.2)	**0.033**
oxLDL (U/mmol LDLc), md (IQR)	15.5 (6.2–17.1)	11.9 (8.8–14.5)	0.569
LDLc (mM), md (IQR)	3.03 (1.90–3.44)	2.40 (1.82–3.1)	0.468
*Intraplaque inflammation*	**SUVmax ≥ 2.85 g/mL (n = 26)**	**SUVmax < 2.85 g/mL (n = 38)**	*p*
LDL(−) (%), md (IQR)	7.2 (5.4–10.9)	8.0 (6.2–10.2)	0.642
oxLDL (U/mmol LDLc), md (IQR)	15.9 (10.1–18.7)	9.9 (7.0–16.9)	0.127
LDLc (mM), md (IQR)	2.26 (1.89–3.18)	2.52 (1.75–3.37)	0.603

oxLDL (oxidized low-density lipoprotein); LDL(−) (electronegative LDL); LDLc (low-density lipoprotein cholesterol); SUVmax (maximal standardized uptake value). The Wilcoxon rank-sum test was used to compare groups; *p* < 0.05 indicates significant differences.

**Table 3 antioxidants-12-00438-t003:** Multivariable logistic regression analysis of predictors of carotid vulnerability.

	OR	95% CI	*p*
(**a**) Plaque stenosis ≥50%			
Proportion of LDL(−) (×1 increase in the logarithm)	5.40	1.15–25.44	**0.033**
Hypertension	0.12	0.01–1.02	0.052
(**b**) Predominantly hypoechoic plaque			
Proportion of LDL(−) (×1 increase in the logarithm)	7.52	1.26–44.83	**0.027**
Prior statin therapy	0.23	0.06–0.79	0.020
(**c**) Diffuse intraplaque neovascularization			
Proportion of LDL(−) (×1 increase in the logarithm)	10.77	1.21–95.93	**0.033**

## Data Availability

All data are contained within the article or as Appendix A.

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
