# Peer review of "Electronegative LDL Is Associated with Plaque Vulnerability in Patients with Ischemic Stroke and Carotid Atherosclerosis"

_antioxidants, 2023, doi:10.3390/antiox12020438_

Round 1
Reviewer 1 Report
This is a very interesting paper. My only concerned is that the author do not discuss why the patient with >90% stenosis are not different than the control group. Also the figure 3 will benefit to have the panel (b) removed.
Author Response
This is a very interesting paper. My only concerned is that the author do not discuss why the patient with >90% stenosis are not different than the control group. Also the figure 3 will benefit to have the panel (b) removed.
We thank the reviewer for his/her kind comments.
We think that the main reason behind the lack of differences between this stenosis group and the control group is the small number of patients in this subgroup (n=10), which explains in part the great variability (as you can see in the box plot). Alternatively, this subgroup included also near-occlusions of the internal carotid artery, a scenario where is radiologically challenged to measure 18FDG uptake (due to the narrowing of the vessel in the context of hemodynamic insufficiency) and may increase also the variability in the measures.
As suggested by the reviewer, Figure 3 (b) has been removed.
Reviewer 2 Report
This manuscript is of interest; however, more biochemical assays are required for supporting the conclusion of this study as listed below:
1. As authors claimed that electronegative LDL is “associated” with the plaque vulnerability in patients with ischemic stroke and carotid atherosclerosis. The causal relationship between electronegative LDL and functions of vascular cells, at least, should be investigated in cell models. For example, the studies regarding EC dysfunction, such as NO production, eNOS activity and monocyte adhesion assay.
2. The levels of lipid oxidation in plasma should be examined.
3. The antioxidant capacity in plasma should be examined.
4. What is the effect of electronegative LDL on the regulation of redox status in ECs? Authors should experimentally address this review point.
5. How electronegative LDL affects the plaque vulnerability, at least, should be discussed in this manuscript? For example, the role of ROS or antioxidants on the expression or activity of MMPs in various vascular cells should be addressed in the revision.
Reviewer 3 Report
The present manuscript described that LDL(-) is related to stenosis and vulnerable atherosclerotic plaque. The methods are well performed and explained but, there are some questions that might be improved by the authors. First of all, the authors comment that the previous treatments are recorded, but I missed the information if the patients with hypercholesterolemia had treatment or not and if this treatment might be masking the results. There are no statically differences significant between dyslipidemia in stenosis groups, but, are there patients with dyslipidemia treatment in any group?
Why did the authors collect blood samples at day 7? Are there some reasons for this timepoint? Have the authors quantified the cholesterol levels at a basal time or at another timepoint different to 7th day?
During the anti-atherogenic properties of HDL study, there are no defined stenosis groups, while in the lipid profile and modified LDL comparison the authors compare the patients’ vs control groups and the stenosis-patient groups vs control group. Have the authors compared the stenosis group in the PAF-AH, PON-1 activity and Che? I suggest adding this comparison. Additionally, when there are more than two groups, the test that is most appropriate is the ANOVA test, not the t-test student. I suggest performing the correct statistical test in each comparison. Moreover, if the authors are separating the patients into different groups, such as the stenosis group (<50%, 50-69%, etc.), they must add the number of patients in each group in the text or in the figure.
Author Response
Please, see the attachment

Round 2
Reviewer 2 Report
Authors have addressed my comments.
Author Response
Thank you for your kind answer